# Simulation of Lanthanum Purification Using a Finite Element Method

**DOI:** 10.3390/ma15093183

**Published:** 2022-04-28

**Authors:** Dehong Chen, Chuang Yu, Zhiqiang Wang, Xiaowei Zhang, Wenli Lu, Dongwei Zhang

**Affiliations:** 1National Engineering Research Center for Rare Earth Materials, GRINM Group Co., Ltd., Beijing 100088, China; chen-dh@126.com (D.C.); wzq97122@126.com (Z.W.); 0420295@163.com (X.Z.); lonely1304178568@163.com (W.L.); zdw2408268511@163.com (D.Z.); 2GRIREM Advanced Materials Co., Ltd., Beijing 100088, China

**Keywords:** lanthanum, zone refining, finite element analysis, the impurity distribution, ultimate purification

## Abstract

The zone refining technology is considered to be one of the most effective means of purifying lanthanum. However, it is tough to obtain the temperature distribution of the molten region through experimental methods. In this study, finite element analysis was used to establish the zone refining simulation model, and the impurity distribution of lanthanum after purification was investigated experimentally. Good agreement between the simulated and experimental results was obtained. The effects of the current and the frequency on the temperature distribution and the width of the region were studied using the simulation model. Through the zone refining experiment, the impurity distributions under different widths of molten region were revealed. Finally, the influence of molten region width on the limiting distribution was calculated by solving the limiting distribution equation.

## 1. Introduction

Rare-earth elements are indispensable resources in science and technology, which are called “industrial vitamins” [1,2,3,4]. Among them, lanthanum is the second most abundant. In recent years, it has been widely used in various applications, such as hydrogen storage materials, filter materials, and energy storage materials, because of its unique physical and chemical properties [5,6,7]. With the rapid development of different industries, requirements for the performance of materials have also increased accordingly. The purity of lanthanum plays a decisive role in its performance. However, due to its active chemical properties and low vapor pressure (10^−4^ Pa at 1500 K), it is more difficult to improve its purity compared with other rare earth metals [8].

Zone refining technology is considered to be one of the most effective means of improving purify. By using the different solubilities of impurities in the solid and liquid phases, directional solidification is carried out to remove impurities. The equilibrium partition coefficient (k_0_) is defined as k_0_ = C_S_/C_L_, where C_S_ and C_L_ represent the solubilities of impurities in the solid phase and liquid phase, respectively. During the zone refining process, impurities of k_0_ < 1 will move to the end region of the metal, following the zone refining direction, but impurities of k_0_ > 1 will move in the opposite direction [9,10,11]. Domestic and foreign scholars have done a lot of research on zone refining technology. D.S. Prasad studied the relationship between zone refining rate and purification efficiency, and found the main impurities—such as Fe and Ni—could be effectively removed [12]. Xinyan Zhai found that Fe in aluminum was easier to remove than Si [13]. Weisheng Liu observed that the thickness of diffusion boundary layer δ during zone refining is closely related to the flow state of the melt [14]. Jun Liu investigated the process of solidification and found that impurities were concentrated at the solidification interface [15]. Huan Zhang researched the relationship between zone refining rate and refining efficiency in the process of purifying tin [16]. Huang Jun used induction heating technology to purify industrial cerium [17]. A significant amount of work has also been done on the finite element method [18,19]. Cheung combined a numerical model and genetic algorithm to establish an optimized melting zone length model to achieve maximum purification efficiency [20]. Tan Yang found that semicircular canals are more suitable than square grooves for zone refining by analyzing finite element simulation data and experimental results [21]. Chen applied the finite element analysis method to simulate the flow field distribution of a refining region [22]. Despite considerable research in the field, the temperature distribution and effective control of the width of the molten region remain obscure. The technique of zone refining is usually carried out by induction heating, and the melting of the metal is the result of an electromagnetic field interaction [23,24,25]. It is tough to obtain the temperature distribution of the molten region during zone refining through experimental methods. There is also a shortage of validated measurement methods for determining the width of an effective molten zone.

In this paper, finite element analysis is used to model the process of purifying lanthanum. The effects of the current and the current frequency on the temperature distribution and the width of the zone are explored. Through the zone melting experiment, the impurity distributions under different widths of molten zone are revealed. Finally, the influence of molten zone width on the ultimate purification is calculated by solving the limiting distribution equation. The present study differs from prior studies in important ways. We report for the first time the relationship between the width of the molten region and AC current. We also reveal the impurity distribution under different widths of molten zone, which has great significance for the actual melting process.

## 2. Simulation Model

### 2.1. Reactor Geometry

Zone refining is based on the principle of electromagnetic induction. When the AC current passes through the water-cooled coil, a transient magnetic field is generated, and the metal placed inside the coil will induce an eddy current. Due to the Joule effect, metals absorb a lot of heat and liquify. While most heat produced by eddy current is absorbed by the metal, causing the temperature to rise, some heat is lost because of conduction and radiation. Considering the symmetry of the zone refining reactor, a two-dimensional axisymmetric model was built for the simulation, as shown in Figure 1. The specific dimensions and boundary conditions of the reactor are listed in Table 1.

### 2.2. Governing Equations

The lanthanum is melted via induction heating. Meanwhile, the current is focused on the metal surface due to the skin effect, which results in uneven temperature distribution in the metal. For both current and frequency changes, the temperature distribution of the molten zone is also changed accordingly. In order to simulate the temperature distribution of the molten phase, the Maxwell equations and heat conduction equations should be solved simultaneously in the transient state. We made the following assumptions in the simulation model for the sake of efficiency:As current thermal effect produces a significant amount of heart, the temperature change caused by phase transition and the influence of crucible can be neglected after the furnace reaches a stable temperature.The whole zone refining process was carried out in vacuum environment. Heat is transferred by conduction and radiation, and the convection is essentially negligible.Under the surface tension of the melt, the shape and length of molten zone do not vary over time. The effect of Lorentz force and gravity on the model was ignored.

The heat conduction equation, the radiation equation, and Maxwell’s equations are given in Table 2 [26].

The heat conduction equation of the reactor can be expressed by Equation (1):(1)∇(k⋅∇T)+q=0

*K* represents thermal conductivity of lanthanum. The relationship between thermal conductivity and temperature is as follows.
*K* = 3.248059 + 0.03346795 × *T* − 2.471847 × 10^−5^ × *T*^2^ + 9.863396 × 10^−9^ × T^3^(2)
*q* is defined as the formation heat per unit volume, which can be calculated by Equation (3):(3)$$q={\left(dV/dx\right)}^{2}\sigma \left(T\right)$$
dV/dx represents voltage gradient per unit length, and $\sigma \left(T\right)$ represents conductivity of lanthanum.

The heat caused by radiation can be calculated by Equation (4):(4)$$Q_{\mathrm{ra}}=\frac{\xi_{1}\xi_{0}}{\xi_{0}+\xi_{1}\left(1-\xi_{0}\right)R/R_{\mathrm{wall}}}\sigma \left(T_{L}^{4}-T_{A}^{4}\right)$$
where σ is the Stefan-Boltzmann constant, 5669 × 10^−8^ W/cm^2^ K^−4^, $\xi_{1}$ represents the radiation coefficient of lanthanum, $\xi_{0}$ represents radiation coefficient of air, and $T_{L}^{}$ and $T_{A}^{}$ are the temperatures of lanthanum and air, respectively. R is the radius of lanthanum, and $R_{\mathrm{wall}}$ is the width of the air region.

The Maxwell equations can be expressed by Equation (5)–(8):(5)∇×H=(σ(T)+iωε)E
(6)∇×E=−iωμH
(7)∇⋅E=0
(8)∇⋅H=0
where *E* represents the electric field intensity, *H* is the magnetic induction intensity, ε is the dielectric permeability, ω represents the pulsation, and μ is the permeability. The numerical values of the parameters used in the simulation are given in Table 2.

### 2.3. Numerical Method

The numerical simulations shown in this paper were conducted using the commercial COMSOL Multiphysics finite element software, and the physical parameters of the material can be found in Table 2. The grid division of the computational domain has a profound effect on the precision of any results. Considering the symmetry of the reactor and the limitation of the grid node structure, the triangular mesh was utilized for computational mesh generation, as shown in Figure 2. After meshing, the total number of grids was 6283, and the mesh was generated with a maximum element size of 1 × 10^−2^ mm and a minimum size of 7.86 × 10^−5^ mm. Computations of skewness (maximum 0.61) and orthogonal quality (minimum 0.32) indicated an acceptable quality of the triangular mesh.

## 3. Experimental Model

As shown in Figure 3, the zone refining reactor was mainly constituted of a furnace, a drive motor, a vacuum pump, alternating current, a crucible, and a coil. To prevent oxidation, the experiment was carried out under vacuum conditions, and the vacuum of furnace was set to 0.01 MPa with the vacuum pump. The coil was placed onto the crucible while keeping the coil position fixed. Additionally, the drive motor connected with the crucible was used to control the movement of the molten zone. As the crucible moved, the heating part also changed. The movement rate was set to 20 mm/min. In addition, the coil was connected to the AC power for efficient induction heating.

The raw materials used for the experiment were lanthanum produced by molten salt electrolysis from LeShan Grinm Advanced materials, which was processed into a round bar with a length of L = 20 cm and a radius of 3 cm and placed in the crucible. Inductively coupled plasma atomic emission spectrometry was used to analyze lanthanum composition. Its specific impurity content is shown in Table 3. It can be seen in the Table 3 that Fe (787 ppm) and Si (345 ppm) were the main impurities in lanthanum, which needed special attention in the experiment. The other impurities (Cu, Cr, Mn, Zn, Ti, Pr, Ce, Co) were close to limiting distribution due to their low contents (<50 ppm), which could be neglected, and the zone melting method could have little effect on their purification.

## 4. Results and Discussion

### 4.1. Simulation Model Validation

The calculated temperature of point 1 (as shown in Figure 1) was simulated independently for the *f* = 30 KHz and *I_coil_* = 55~75 A conditions by COMSOL to confirm the validity of the simulation model. Additionally, the calculated temperature ($T_{i-Cal}$) was compared to the experimental temperature ($T_{i-Exp}$) which was measured by using a double color infrared thermometer under the same conditions. As shown in Figure 4, the calculated temperature of point 1 was close to the experimental temperature when *I_coil_* was controlled in the range of 60–70A. In 70 A, the calculated temperature was in very good agreement with the experimental temperature. By comparing five temperatures for both calculated and experimental temperatures, the average relative error ($S^{*}=\pm \frac{100}{n}\overset{n}{\underset{i=1}{\sum }}\left|\frac{T_{i-Cal}-T_{i-Exp}}{T_{i-Exp}}\right|$) was used to evaluate the proximity between the calculated temperature and the experimental temperature [27]. We then computed the relative error of the temperature as 3.64%, which demonstrated the accuracy and reliability of the simulation model.

### 4.2. Influence of Frequency on Radial Temperature Distribution

Figure 5 represents the temperature distribution of the zone refining reactor at *I_coil_* = 70 A and *f* = 30 KHz. As shown in the Figure 5, the surface temperature of the melting zone was 1020 °C, and the calculated temperature was in very good agreement with the experimental temperature. Affected by eddying effect and joule effect, the temperature of the molten region was very high. However, the temperature of the coil was close to room temperature due to cooling water. Thermal radiation has a significant promoting effect on the increasing of air temperature. It is visible in Figure 5 that there was a temperature difference between the surface and the center of the molten zone. The eddy current usually exists on the surface of the molten region owing to the skin effect, and leads to higher surface temperature than the central temperature. The AC frequency is the main factor that affects skin depth, and it has a dominant role in the radial temperature. Therefore, the heating depth can be effectively controlled by selecting an appropriate frequency.

Figure 6 shows the radial temperature distribution of the molten region at *I_coil_* = 75 A and *f* = 20, 30, and 40 KHz. It can be seen in Figure 6 that the temperature increased gradually from the center point to the surface, and the temperature at the center was the lowest in the molten region. With the decrease in AC frequency, the center temperature of molten zone decreased obviously. The center temperature was 1030 °C when the frequency was 40 KHz, and it was 981 °C when the frequency reduced to 30 KHz, which is only 61 °C higher than the melting point of lanthanum. However, the central temperature was only 899 °C when the frequency decreased to 20 KHz, which is significantly lower than the melting point of lanthanum. At this time, unmelted inclusions appeared in the molten region, which are harmful to the migration of impurities. With the increase in current frequency, the difference between surface temperature and center temperature also gradually increased.

### 4.3. Influence of Current on Axial Temperature Distribution

In order to prevent the temperature of the molten zone from being too low because of a small current, the current range of 70–81 A was selected for research. Figure 7 displays the axial temperature distribution of molten region at *f* = 30 KHz and *I_coil_* = 70, 71, 73, 75, 77, 79, and 81 A. It can be observed from Figure 7 that the temperature distribution inside the molten zone gradually decreased from the center point to the sides, which was due to the induced magnetic field being mainly concentrated at the center of the coil and resulting in a large induced current in this region. The center temperature was 928 °C at 70 A, and it increased to 1043 °C, which is much higher than the melting point of lanthanum, when the current increased to 81 A. Regarding the area with a temperature greater than 920 °C as the valid molten region, the width of valid molten region gradually increased with the increase in current.

### 4.4. Influences of Frequency and Current on Molten Region Width

Figure 8 shows the area of the valid molten zone at *f* = 30 KHz and *I_coil_* = 70, 71, 75, and 81 A. We can observe from Figure 8 that the valid molten zone area increased with the increase in current. *Z* represents the width of the valid molten region. When the current was 70 A, *Z* was only 2 cm, but it increased to 4 and 6 cm when the current increased to 75 and 81 A respectively. The value of *Z* is the result of current magnitude and frequency. Therefore, the influence of current magnitude and frequency must be taken into account when the width of the valid molten zone is calculated.

Figure 9 shows the width of the molten zone when the frequency was 20, 30 and 40 KHz, and the current increased from 71 to 81 A. The width of molten region *Z* = 0 at *f* = 20 KHz and *I_coil_* increased from 71 to 77 A. When the frequency was small, the center temperature of the molten zone was lower than the melting point of the lanthanum. Some areas were still solid, and impurities could not migrate effectively. A significant increase in *Z* (from 2 to 6 cm) was observed when the AC frequency was 30 kHz and the current increased from 71 to 77 A. However, when the AC frequency was 40 kHz, the improvements in Z were clearly slowing down, as the current only increased from 4.5 to 7 cm. Therefore, the width of the molten zone can be effectively controlled to achieve accurate heating by controlling the current frequency and size.

### 4.5. Influence of the Width of the Molten Region on Impurity Distribution

To further investigate how the width of the molten region affects impurity distribution, we conducted a zone refining experiment. The relationships between the width of the molten region and the frequency and current were studied in Section 4.1. In order to improve the efficiency of zone melting and ensure that the experimental equipment would not be damaged by excessive power current, the current frequency was selected as 30 KHz for the experimental work. In the zone refining experiment, zone refining rate was 20 mm/h, and the current was set to 71, 75, or 81 A to ensure the molten zone width was 2, 4, or 6 cm. After ten passes of zone refining, the distribution of impurity concentration was obtained after purification by ICP-AES.

Figure 10 and Figure 11 represent the concentration distributions of Fe and Si in the metal after zone refining under different widths of molten region. It can be observed in Figure 10 and Figure 11 that the concentrations of impurities in the metal gradually increased from the beginning to the end after zone refining. When the width of molten region was 2 cm, the concentrations of Fe and Si in the beginning were 152 and 45 ppm, respectively, much lower than in the raw material. At the end of the process, Fe rapidly increased to 2025 ppm, and Si increased to 528 ppm at the same time. This happened because the equilibrium distribution coefficients of Fe and Si are less than 1, meaning such impurities are more likely to exist in the liquid phase during the melting process. As the experiment progressed, impurities also followed the molten region from the beginning to the tail, and were enriched in the tail. With the increase in the width of the molten region, the impurity concentration increased gradually in the front of the sample, and the purification effect was weakened. When the width of the molten zone was too large, the impurity could not be diffused due to the decrease in temperature, which could not be effectively migrated, resulting in the weaken of purification. Therefore, a lower width of the molten zone should be selected to improve the purification efficiency in the zone refining process.

### 4.6. Effect of Width of Molten Region on Limiting Distribution

After long-term theoretical practice, it has been found that impurities cannot be completely removed, and the concentration will reach a relatively stable distribution after multiple zone refining, which is called the limiting distribution. During zone refining, there are two solid–liquid interfaces, the solidification interface and the melting interface. When k_0_ < 1, impurities enter the liquid phase through the melting interface, move from the beginning to the end, and finally move out of the molten region through the solidification interface. At the same time, some tail impurities will be re-entered into the molten region through the melting interface, and spread throughout the whole region through the molten region mixing. This phenomenon is called backflow, as shown in Figure 12, which inhibits the purification.

In the previous zone refining, the segregation effect played a leading role in impurity migration because the impurity concentration at the end of the process was low; there was little impurity backflow introduced into the molten region. However, as the zone refining time increased, the impurity concentration increased in the direction of zone refining, and the impurity concentration in the tail increased sharply. The inhibition of the purification of the segregation interface also increased. After several rounds of zone refining, the impurity mass that moves out of the molten region through the solidification interface is equal to that entering the region through the melting interface. The distribution of impurities in the metal does not change with any further increase in zone melting times. At this time, the impurity reaches the limiting distribution state.

The limiting distribution equation has been given by Pfann [28,29,30], as shown in Equation (9).
(9)$$C_{s}(x)=Ae^{Bx}$$
where *A* and *B* are constants. The relationship between constant *B* and the equilibrium partition coefficient is given by Equation (10). The values of *A* and *B* can be obtained by solving the equilibrium distribution coefficient.
(10)$$k_{0}=\frac{Bz}{e^{Bz}-1}$$
(11)$$A=\frac{C_{0}BL}{e^{BL}-1}$$

Additionally, the relationships between *A* and *B* in Fe and Si, and the width of molten zone, were calculated, as shown in Table 4.

Figure 13 and Figure 14 show the limiting distributions of Fe and Si under different molten zone widths, respectively. It can be seen in each figure that the limiting distribution concentrations of Fe and Si were lower at the beginning and higher at the end, which is consistent with the previous experimental phenomena. In addition, as the width of the molten zone decreased, the limiting distribution concentration also correspondingly decreased. It favored the maximization of purification. When the molten region width is small, the impurity concentration that the molten region can accommodate is limited, which has a certain mitigation effect on the backflow of impurities. Therefore, the width of the molten zone should be controlled within a small range in actual production processes, in order to improve purity.

## 5. Conclusions

This paper analyzed the influences of different molten zone widths on the impurity distribution by establishing a simulation model of zone refining to research the relationships among the molten region width, temperature distribution, frequency, and the magnitude of current. The conclusions are as follows:We compared the calculation results and the experimental results. The average relative error was 3.64%, which indicates that the simulation model is accurate and reliable.The radial temperature distribution of the molten region increased from the center to the surface. With the increase in the frequency, the temperature at the center increased, and the temperature difference between the surface and the center increased gradually.With the increase in the width of the molten zone, the impurity concentration increased gradually in the front of the sample, and the purification effect was weakened. After 10 rounds of zone refining and holding the width of molten at 2 cm, the concentrations of Fe and Si at the beginning were 152 and 45 ppm respectively, which are much lower concentrations than those in the raw material, indicating that Fe and Si can be efficiently removed.As the width of the molten region decreased, the limiting distributions of impurities also correspondingly decreased, favoring the maximization of purification effect.

## Figures and Tables

**Figure 1 materials-15-03183-f001:**
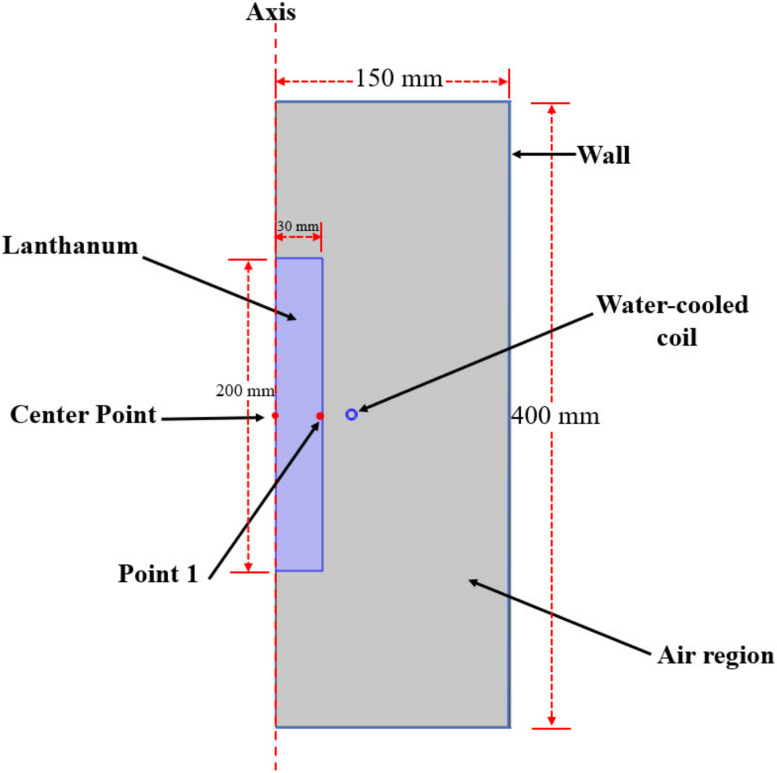
Geometrical model of the zone refining reactor.

**Figure 2 materials-15-03183-f002:**
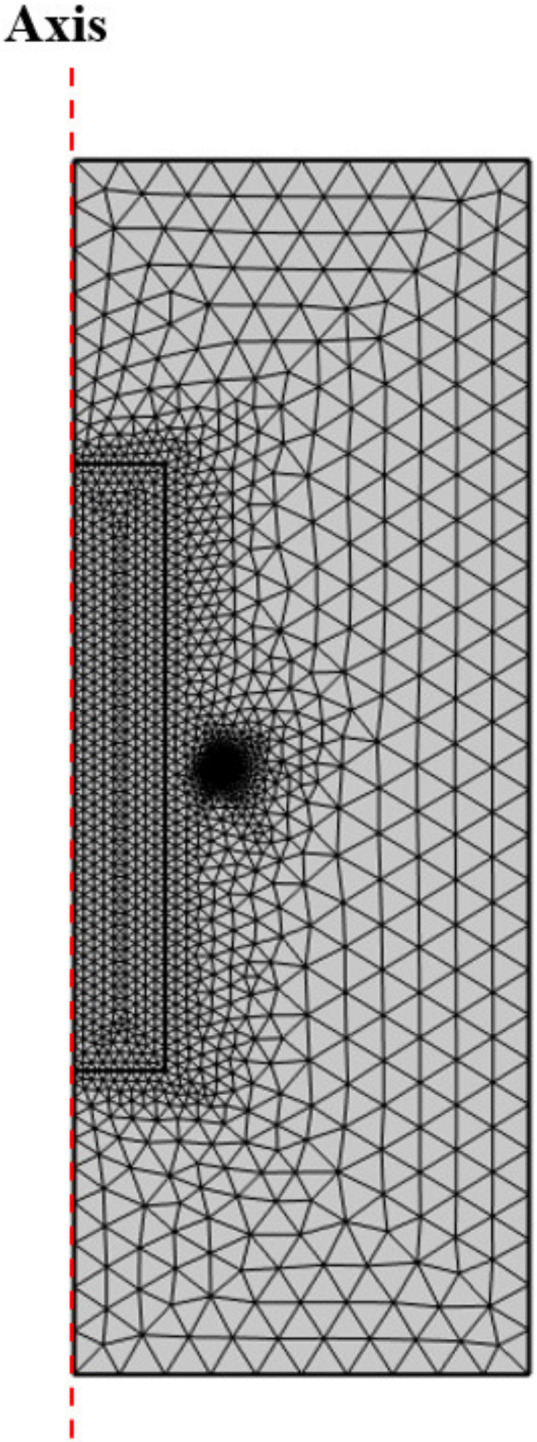
Zone refining reactor meshing.

**Figure 3 materials-15-03183-f003:**
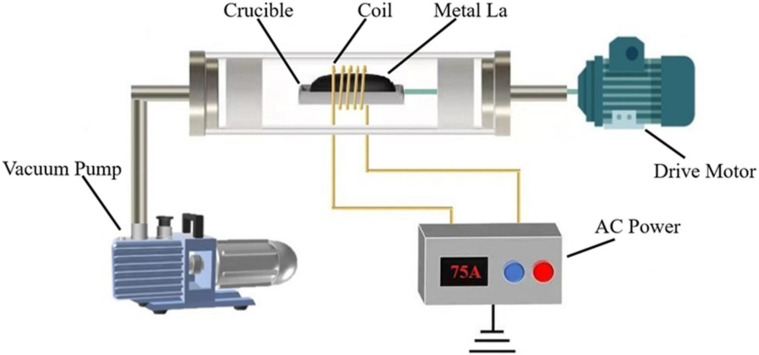
Schematic of the zone refining reactor.

**Figure 4 materials-15-03183-f004:**
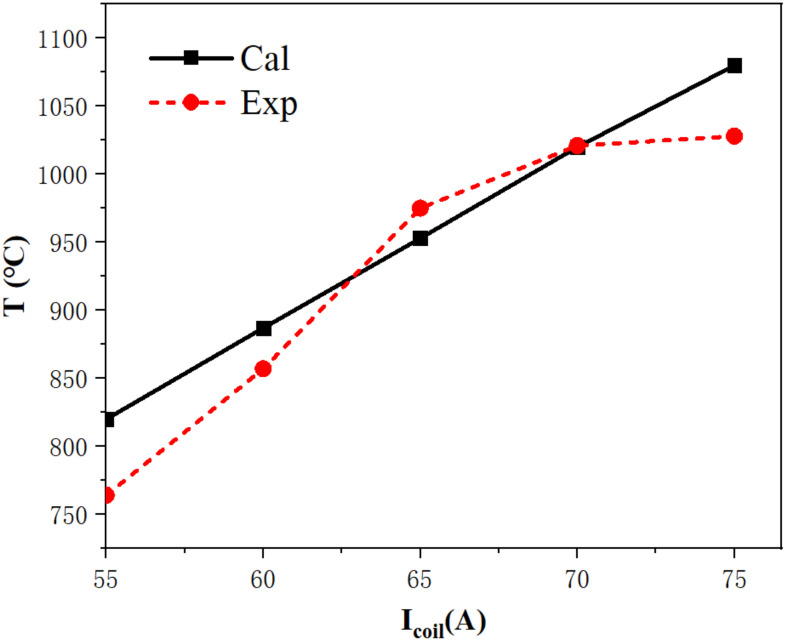
The calculated and experimental temperature of point 1.

**Figure 5 materials-15-03183-f005:**
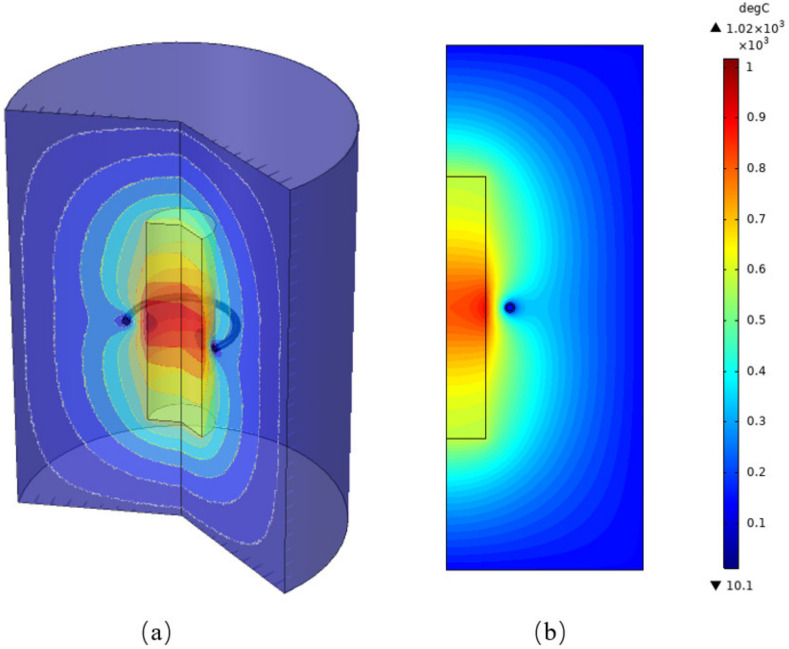
The temperature profile of the zone melting reactor. (**a**) 3D; (**b**) 2D.

**Figure 6 materials-15-03183-f006:**
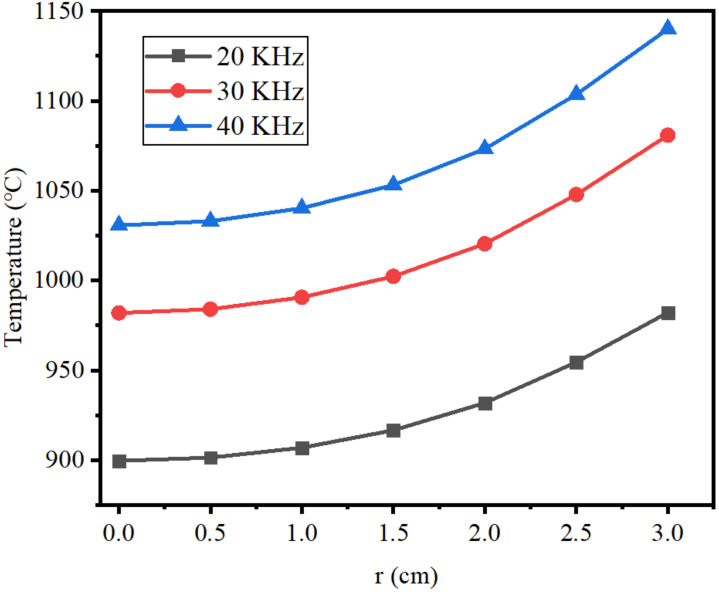
Radial temperature distribution of molten region.

**Figure 7 materials-15-03183-f007:**
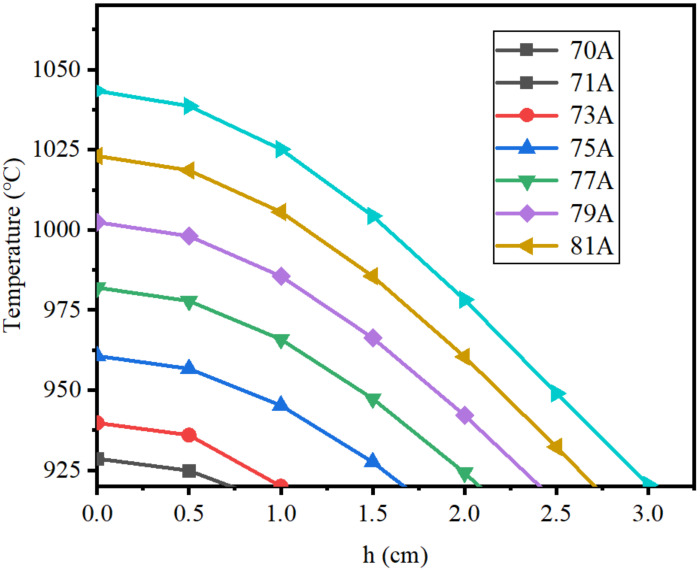
Axial temperature distribution of the molten region.

**Figure 8 materials-15-03183-f008:**
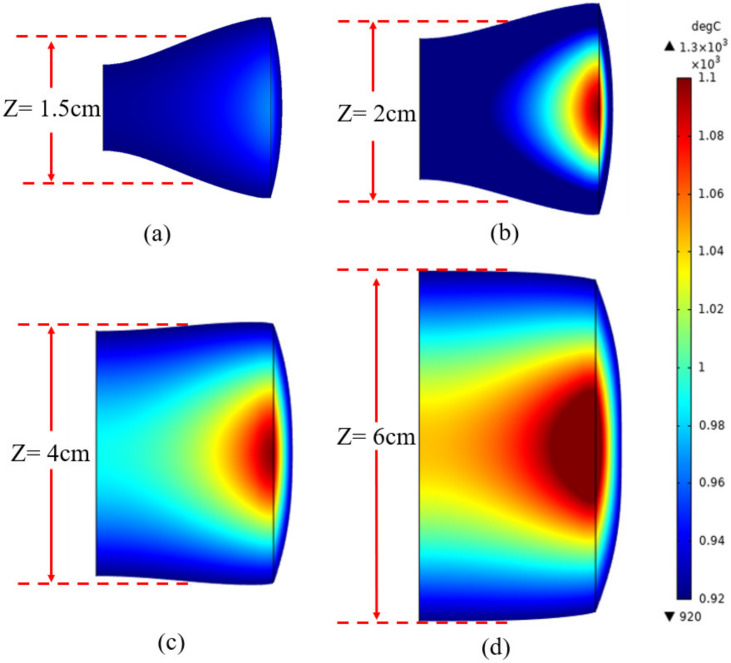
The area of the valid molten region under different currents. (**a**) *I_coil_* = 70 A; (**b**) *I_coil_* = 71 A; (**c**) *I_coil_* = 75 A; (**d**) *I_coil_* = 81 A.

**Figure 9 materials-15-03183-f009:**
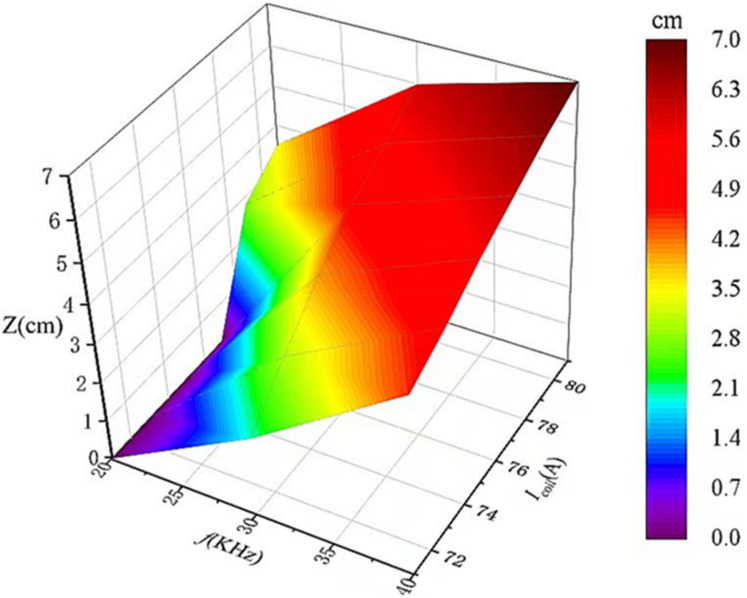
The variations in *Z* with current and frequency.

**Figure 10 materials-15-03183-f010:**
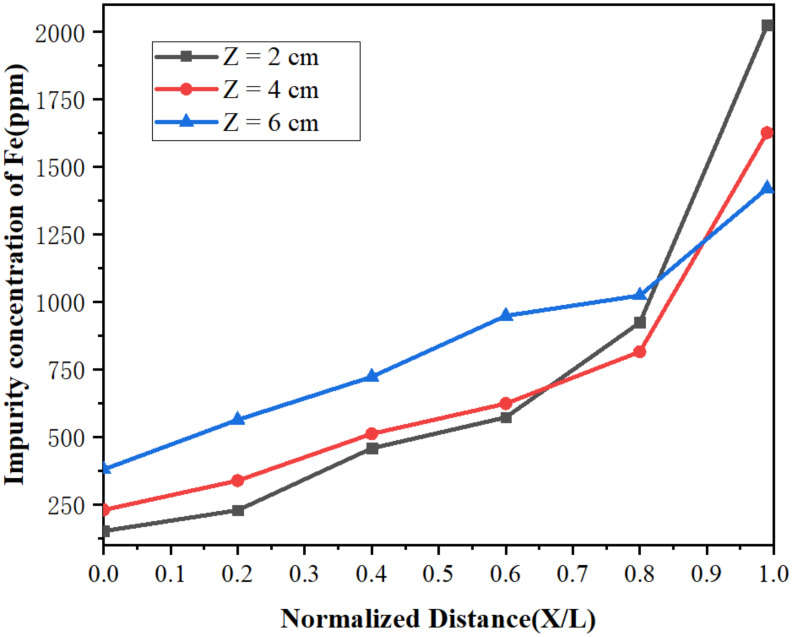
Distribution of Fe under different widths of molten region.

**Figure 11 materials-15-03183-f011:**
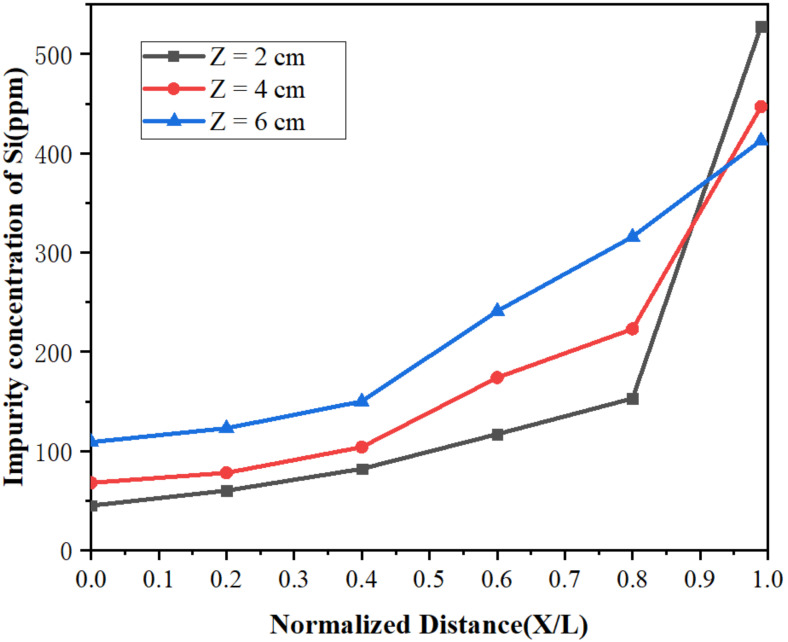
Distribution of Si under different widths of molten region.

**Figure 12 materials-15-03183-f012:**
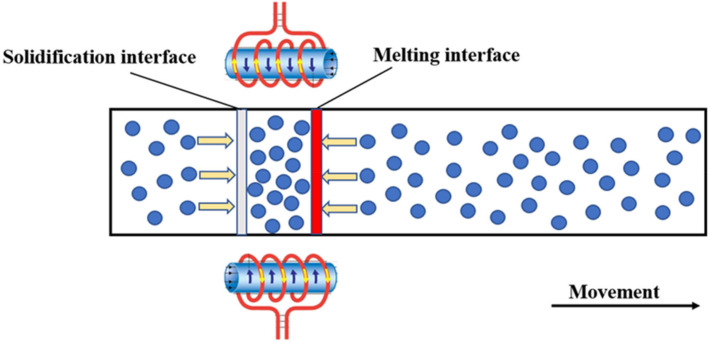
Impurity backflow diagram.

**Figure 13 materials-15-03183-f013:**
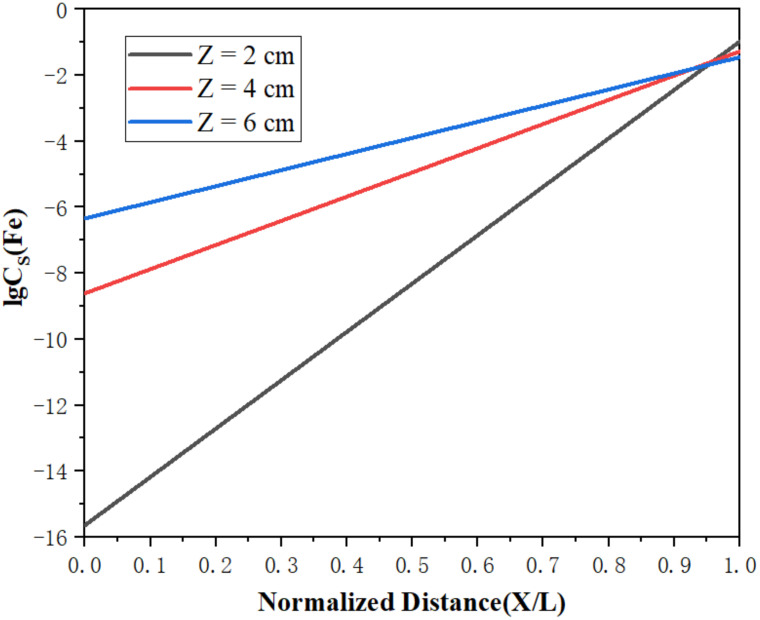
Limiting distribution of Fe.

**Figure 14 materials-15-03183-f014:**
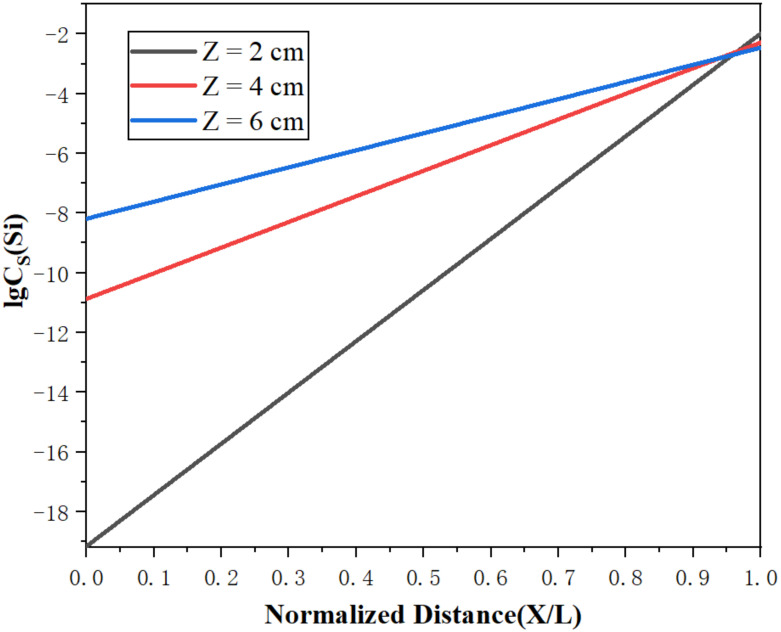
Limiting distribution of Si.

**Table 1 materials-15-03183-t001:** Dimensions and boundary conditions of the zone refining reactor.

Region	Items	Data
Water-cooled coil	Outer diameter	7 mm
	Inner diameter	5 mm
Air region	Width	150 mm
	Length	400 mm
Lanthanum	Width	30 mm
	Length	200 mm
Boundary conditions	Temperature of cooling water	283 K
	Temperature of reactor wall	293 K
	Cooling water mass flow rate	1 kg/min
	Emissivity of metal	0.5

**Table 2 materials-15-03183-t002:** The numerical values of the parameters.

Parameter	Value	Unit
μ	1	H/m
$\sigma \left(T\right)$	0.2	MS/cm
$\xi_{1}$	0.64	1
$\xi_{0}$	0.5	1
ε	1	1

**Table 3 materials-15-03183-t003:** The impurity content of lanthanum.

Impurity	Fe	Cu	Cr	Si	Mn
(ppm)	787	3	6	345	11
**Impurity**	Zn	Ti	Pr	Ce	Co
(ppm)	33	25	8	3.37	2

**Table 4 materials-15-03183-t004:** The relationships between A and B and the width of the molten zone.

Width	Fe		Si	
	*A*	*B*	*A*	*B*
Z = 2 cm	2.25 × 10^−16^	33.79	6.94 × 10^−20^	39.55
Z = 4 cm	2.45 × 10^−9^	16.89	1.35 × 10^−11^	19.77
Z = 6 cm	4.56 × 10^−7^	11.26	6.56 × 10^−9^	13.18

## Data Availability

Data sharing is not applicable to this article.

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
