# Peer review of "Simulation of Lanthanum Purification Using a Finite Element Method"

_materials, 2022, doi:10.3390/ma15093183_

Round 1
Reviewer 1 Report
I have reviewed the manuscript titled “Simulation and experimental research on purification of lan thanum by zone refining”. Author(s) have used the finite element method to establish the zone refining simulation model and to establish the relationship between the current and the frequency on the temperature distribution and the width of the region. The experimental method is used by them to measure the impurity distribution of lanthanum after purification.
The topic is interesting for the research community. The manuscript deserves to be published. However, there are some minor issues that should be taken care of before publication:
- The submitted manuscript needs language editing as the manuscript has some grammatical and typographical errors. For example ‘which indicated that the simulation model was accuracy and reliability’ (Line 129-130)
- The meaning of any symbol used should be given first, for example, k0=CS/CL in introduction section. Also, MS has mentioned ‘k’ also (line 227) with no meaning.
- Author(s) are advised to add some more numerical work related to topic.
- Author(s) have used finite element method based COMSOL Multiphysics finite element software. They are advised to add one more paragraph for COMSOL code details for zone refining simulation model, meshing and formulation in section 2.3 Numerical method.
- Author(s) should explain the location of temperature of point 1 (line 120)
Author Response
Dear Reviewer:
Thank you for your letter and comments concerning our manuscript entitled “Simulation and experimental research on purification of lan-thanum by zone refining” (ID: materials-1643748). Those comments are all valuable and very helpful for revising and improving our paper, as well as the important guiding significance to our researches. We have studied comments carefully and have made correction which we hope meet with approval. Revised portion are marked in red in the paper. The main corrections in the paper and the responds to the reviewer’s comments are as flowing:
Point 1: The submitted manuscript needs language editing as the manuscript has some grammatical and typographical errors. For example ‘which indicated that the simulation model was accuracy and reliability’ (Line 129-130)
Response 1: Thank you for your valuable and thoughtful comments. We have carefully checked and improved the English writing in the revised manuscript. Such as:
(which demonstrated the accuracy and reliability of the simulation model.)
(The relationship between the current and the frequency on the temperature distribution and the width of the region were studied using the simulation model.)
(In recent years, it has been widely used in various fields such as hydrogen storage materials, filter materials, and energy storage materials because of its unique physical and chemical properties)
(A significant amount of work have also been done on the finite element method.)
Point 2: The meaning of any symbol used should be given first, for example, k0=CS/CL in introduction section. Also, MS has mentioned ‘k’ also (line 227) with no meaning.
Response 2: We have added explanations to all the symbols in this paper.
(The equilibrium partition coefficient (k0) is defined as k0=CS/CL, CS and CL represent the solubility ratio of impurities in solid phase and liquid phase, respectively.)
(When k0 < 1, impurities enter the liquid phase through the melting interface, move from the beginning to the end, and finally move out of the molten region through the solidification interface.)
(The heat conduction equation of the reactor can be expressed by equation (1):
(1)
K represents thermal conductivity of lanthanum, the relationship between thermal conductivity and temperature is as follows:
K=3.248059+0.03346795×T-2.471847×10-5×T2+9.863396×10×T3 (2)
q is defined as the formation heat per unit volume which can be calculated by equation (3):
(3)
represents voltage gradient per unit length, represents conductivity of lanthanum.
The heat caused by radiation can be calculated by equation (4):
(4)
Among them, is the Stefan-Boltzmann constant, 5.669×10−8 W/cm2 K−4, represents radiation coefficient of lanthanum, represents radiation coefficient of air, and are the temperature of lanthanum and air, respectively. and are the radius of lanthanum, are the width of air region.
The Maxwell equations can be expressed by equation (5-8):
(5)
(6)
(7)
(8)
E represents the electric field intensity, H is the magnetic induction intensity, is the dielectric permeability, represents the pulsation, is the permeability. The numerical values of the parameters used in the simulation were given in Table 2.)
Point 3: Author(s) are advised to add some more numerical work related to topic.
Response 3: The recent numerical work have added in the main text.
A significant amount of work has also been done on the finite element method [18-19]. Cheung combined a numerical model and genetic algorithm to establish an optimized melting zone length model to achieve maximum purification efficiency [20]. Tan Yang reported that semicircular canals was more suitable than square grooves for the purifica-tion of zone refining by combining finite element simulation data and experimental re-sults analysis [21]. Chen applied the finite element analysis method to simulate flow field distribution of the refining region [22].
Point 4: Author(s) have used finite element method based COMSOL Multiphysics finite element software. They are advised to add one more paragraph for COMSOL code details for zone refining simulation model, meshing and formulation in section 2.3 Numerical method.
Response 4: The zone refining simulation model and formulation were given in Figure 1, and formulation have also been given in section 2.2. According to your suggestion, zone refining reactor meshing was added.
Figure 2. Zone refining reactor meshing.
Point 5: Author(s) should explain the location of temperature of point 1 (line 120)
Response 5: The location of point 1 was shown in Figure. 1 and has been explained in further detail.
The calculated temperature of point 1 (as shown in Figure. 1) was simulated independently for the f = 30KHz and Icoil= 55~75A conditions by COMSOL to confirm the validity of the simulation model.
Special thanks to you for your good comments.

Reviewer 2 Report
To improve the manuscript, please provide additional information and perform the following changes:
- On page 1, line 30, in the phrase “However, due to its active chemical properties and low vapor pressure…” should be given some quantifiable data for the properties.
- On page 3, referring to the heat conduction equation, radiation equation and Maxwell’s equations given in Table 2 should be provided proper references. Also, the constitutive terms of the equations should be briefly explained even the above equations are well known.
- On page 3, lines 93-95, in the phrase “The numerical simulations shown in this paper was conducted using the commercial COMSOL Multiphysics finite element software, and the physical parameters of material can be found in database.” should be mentioned the physical parameters of material, including their values, that were used in the numerical simulations.
- On page 4, lines 106-107, should be changed “0.01Mpa” with “0.01 MPa”. Referring to the phrase “The drive motor connected with the crucible was used to control the movement of the molten zone.” should be clarified how was controlled the movement of the molten zone, and which parameters and their values were involved in this step.
- On page 4, lines 111-113, should be specified the producer of the raw materials, the type, and purity of the molten salt, and electrolysis conditions, along with the processing parameters and their values to produce a round bar with a length of L = 20 cm and a radius of 3 cm. The round bar should be also analyzed in terms of physical properties and the results should be also presented in the manuscript to compare the input data from the numerical simulations with the ones from the experimental works.
- On page 4, lines 113-114, in the phrase “Inductively coupled plasma atomic emission spectrometry was used to analyze lanthanum composition…” should be mentioned the type of the spectrometer and the measurement conditions. Should be also clarified the zone from which were taken samples for measurements, and how was studied the material homogeneity.
- On page 4, lines 126-127, should be given a reference for the equation showing the average relative error, and the number of considered temperatures (n) used to determine the average value should be also specified even in Figure 3 can be seen 5 temperatures for both calculated and experimental temperatures.
- On page 5, in subsection “4.1.2. Influence of frequency on radial temperature distribution” are given the results for the temperature distribution of the zone refining reactor at Icoil = 75 A and f = 30 KHz. I recommend you to show the results in Figure 4 and Figure 5 with adequate comments for Icoil = 70 A and f = 30 KHz, where the calculated temperature was in very good agreement with the experimental temperature (Figure 3). Figure 5 (Radial temperature distribution of molten region) presents the simulation results for f = 20 KHz, f = 30 KHz, and f = 40 KHz, but in the experimental work was selected f = 30 KHz. In this respect, should be clarified the selection of the frequency f = 30 KHz in the experimental work.
- On page 6, line 161, should be checked “f = 75A” since f is the abbreviation for frequency, and A is the measurement unit for current.
- In the subsection “4.1.3. Influence of current on axial temperature distribution”, the results for Icoil =71A, 73A, 75A, 77A, 79A, 81A are presented but the results for Icoil =70A should be also presented in Figure 6. Clarify why the values of Icoil were selected from 71A to 81A, whereas in subsection "4.1.1. Simulation Model validation" the studied range of Icoil was between 55A and 75A. The range of Icoil should be similar in all the subsections where the influence of Icoil is studied.
- In the subsection “4.1.4. Influence of frequency and current on molten region width”, the results for f = 30 KHz and Icoil = 71A, 75A, 81A are presented but the results for Icoil =70A should be also presented in Figure 7.
- On page 8, lines 198-199, in the phrase “The relationship between the width of the molten region and frequency and current studied in the previous paper” should be mentioned the reference number of the previous paper of the authors. The zone refining rate and the number of passes of zone refining should be also specified in the actual manuscript submitted to Materials because in the actual formulation can be understood that these parameters refer to the previous paper of the authors.
- On page 8, lines 203-204, in the phrase “Figure 9 and Figure 10 represent the concentration distribution of impurity Fe and Si in metal after zone refining under different conditions.” should be specified the conditions (e.g. current, frequency, melting temperature, including their values) instead expressing them in a general way.
- On page 1, lines 34-38, “The equilibrium partition coefficient is defined as k0=CS/CL, which represents the solubility ratio of impurities in solid and liquid phase. During the zone refining process, the impurities of k0<1 will move to the end region of metal following the zone refining direction but impurities of k0 >1 will move along the opposite direction [8-10].”, whereas on page 9, line 227, it is mentioned “k < 1”. The notation for the equilibrium partition coefficient should be checked.
- On page 10, referring to “formula (4-1), (4-2)…”, should be replaced the word "formula" with "equation", and the numbering of the equations should be checked since the equations were noted as (1), (2), …
- In Table 4, in column A for Fe and Si elements should be put as a superscript the number after each ×10 since it is an exponent. However, the results are presented only for the main impurities (Fe and Si) with a higher content (≥ 50 ppm). In the comments should be discussed why the other impurities that were shown in Table 3 (Cu, Cr, Mn, Zn, Ti, Pr, Ce, Co) with a lower content (< 50 ppm) were neglected.
- On page 12, in section “5. Conclusions”, it was stated that “After zone refining, Fe and Si can be efficiently removed” but should be clarified the number of passes of zone refining and the purity, including the type and content of the remained impurities of the final material.
Author Response
Dear Reviewer:
Thank you for your letter and comments concerning our manuscript entitled “Simulation and experimental research on purification of lan-thanum by zone refining” (ID: materials-1643748). Those comments are all valuable and very helpful for revising and improving our paper, as well as the important guiding significance to our researches. We have studied comments carefully and have made correction which we hope meet with approval. Revised portion are marked in red in the paper. The main corrections in the paper and the responds to the reviewer’s comments are as flowing:
Point 1: On page 1, line 30, in the phrase “However, due to its active chemical properties and low vapor pressure…” should be given some quantifiable data for the properties.
Response 1: Thank you for your reminder, and we have added the quantifiable vapor pressure data in introduction. Readers can find specific data in the reference [8].
(However, due to its active chemical properties and low vapor pressure(10-4Pa at 1500K), it is more difficult to improve its purity compared with other rare earth metals [8].)
Point 2: On page 3, referring to the heat conduction equation, radiation equation and Maxwell’s equations given in Table 2 should be provided proper references. Also, the constitutive terms of the equations should be briefly explained even the above equations are well known.
Response 2: Reference related to equations has been provided in this paper. And we have added explanations to all the symbols in this paper.
[26] Hou, Y. Q. Theoretical analysis and modeling of polysilicon preparation process by improved Siemens method. PhD thesis, Kunming University of Science and Technology, Kunming, 2013.
(The equilibrium partition coefficient (k0) is defined as k0=CS/CL, CS and CL represent the solubility ratio of impurities in solid phase and liquid phase, respectively.)
(When k0 < 1, impurities enter the liquid phase through the melting interface, move from the beginning to the end, and finally move out of the molten region through the solidification interface.)
(The heat conduction equation of the reactor can be expressed by equation (1):
(1)
K represents thermal conductivity of lanthanum, the relationship between thermal conductivity and temperature is as follows:
K=3.248059+0.03346795×T-2.471847×10-5×T2+9.863396×10×T3 (2)
q is defined as the formation heat per unit volume which can be calculated by equation (3):
(3)
represents voltage gradient per unit length, represents conductivity of lanthanum.
The heat caused by radiation can be calculated by equation (4):
(4)
Among them, is the Stefan-Boltzmann constant, 5.669×10−8 W/cm2 K−4, represents radiation coefficient of lanthanum, represents radiation coefficient of air, and are the temperature of lanthanum and air, respectively. and are the radius of lanthanum, are the width of air region.
The Maxwell equations can be expressed by equation (5-8):
(5)
(6)
(7)
(8)
E represents the electric field intensity, H is the magnetic induction intensity, is the dielectric permeability, represents the pulsation, is the permeability. The numerical values of the parameters used in the simulation were given in Table 2.)
Point 3: On page 3, lines 93-95, in the phrase “The numerical simulations shown in this paper was conducted using the commercial COMSOL Multiphysics finite element software, and the physical parameters of material can be found in database.” should be mentioned the physical parameters of material, including their values, that were used in the numerical simulations.
Response 3: I apologize for my negligence again and the physical parameters of material that were used in the numerical simulations have been metioned in Table 2.
(The numerical simulations shown in this paper was conducted using the commercial COMSOL Multiphysics finite element software, and the physical parameters of material can be found in Table2.)
Table 2. The numerical values of the parameters.
|
Parameter |
Value |
Unit |
|
1 |
H/m |
|
|
|
0.2 |
MS/cm |
|
0.64 |
1 |
|
|
0.5 |
1 |
|
|
1 |
1 |
Point 4: On page 4, lines 106-107, should be changed “0.01Mpa” with “0.01 MPa”. Referring to the phrase “The drive motor connected with the crucible was used to control the movement of the molten zone.” should be clarified how was controlled the movement of the molten zone, and which parameters and their values were involved in this step.
Response 4: To make it easier for readers to understand, we have described the experimental part in more detail. The related parameters and their values were involved in this step.
(To prevent oxidation, the experiment was carried out under vacuum conditions, and the vacuum of furnace was set at 0.01 MPa by vacuum pump. The coil was placed onto the crucible while keeping the coil position fixed. And the drive motor connected with the crucible was used to control the movement of the molten zone. As the crucible moves, the hearting part also changes. The movement rate was set at 20 mm/min.)
Point 5: On page 4, lines 111-113, should be specified the producer of the raw materials, the type, and purity of the molten salt, and electrolysis conditions, along with the processing parameters and their values to produce a round bar with a length of L = 20 cm and a radius of 3 cm. The round bar should be also analyzed in terms of physical properties and the results should be also presented in the manuscript to compare the input data from the numerical simulations with the ones from the experimental works.
Response 5: The materials used for the experiment were purchased directly from other companies.
The type of the molten salt、purity、electrolysis conditions and processing parameters are confidential and therefore cannot be available. There is also a lack of effective measurement methods for the physical properties. In addition, there is a lack of effective measurement methods for us to analyzed the round bar.
(The raw materials used for the experiment were lanthanum produced by molten salt electrolysis from LeShan Grinm Advanced materials, which were processed into a round bar with a length of L = 20cm and a radius of 3cm and placed in the crucible.)
Point 6: On page 4, lines 113-114, in the phrase “Inductively coupled plasma atomic emission spectrometry was used to analyze lanthanum composition…” should be mentioned the type of the spectrometer and the measurement conditions. Should be also clarified the zone from which were taken samples for measurements, and how was studied the material homogeneity.
Response 6: The whole composition analysis was entrusted to relevant analysis institutions. We don't know the specific measurement conditions and types, which is not the focus of this paper. And the molten salt electrolysis is regarded as the most uniform method for making metals. Generally, the uniformity of materials produced by molten salt electrolysis does not need to be considered. But your question is very suggestive and deserves our attention in the future.
Point 7: On page 4, lines 126-127, should be given a reference for the equation showing the average relative error, and the number of considered temperatures (n) used to determine the average value should be also specified even in Figure 3 can be seen 5 temperatures for both calculated and experimental temperatures.
Response 7: The relevant reference and the number of considered temperatures have been added.
(By comparing 5 temperatures for both calculated and experimental temperatures, the average relative error () was used to evaluate the proximity between the calculated temperature and the experimental temperature [27].)
- Ya, D. L.; Gang, X. A CFD model for gas uniform distribution in turbulent flow for the production of titanium pigment in chloride process. Chinese Journal of Chemical Engineering 2016, 24(06), 749-756.
Point 8: On page 5, in subsection “4.1.2. Influence of frequency on radial temperature distribution” are given the results for the temperature distribution of the zone refining reactor at Icoil = 75 A and f = 30 KHz. I recommend you to show the results in Figure 4 and Figure 5 with adequate comments for Icoil = 70 A and f = 30 KHz, where the calculated temperature was in very good agreement with the experimental temperature (Figure 3). Figure 5 (Radial temperature distribution of molten region) presents the simulation results for f = 20 KHz, f = 30 KHz, and f = 40 KHz, but in the experimental work was selected f = 30 KHz. In this respect, should be clarified the selection of the frequency f = 30 KHz in the experimental work.
Response 8: Thank you for your careful reading and valuable suggestions. The data in Figure 4 has been replaced with Icoil = 70 A and f = 30 KHz. And the reason for selecting f = 30 KHz has also clarified. The reason why 75A was selected instead of 70A in Figure. 5 is that when the current was 70A and the frequency was 20kHz, the temperature of the molten zone was low and there was an unmelted area, which is not conducive to the migration of subsequent impurities.
(Figure 4 represents the temperature distribution of the zone refining reactor at Icoil = 70A and f = 30KHz. As shown in the figure, the surface temperature of the melting zone was 1020°C, where the calculated temperature was in very good agreement with the experimental temperature.)
Figure 4. The temperature profile of zone melting reactor. (a) 3D; (b) 2D.
(In order to improve the efficiency of zone melting and ensure that the experimental equipment will not be damaged by excessive power current, the current frequency would be selected as 30kHz in in the experimental work.)
Point 9: On page 6, line 161, should be checked “f = 75A” since f is the abbreviation for frequency, and A is the measurement unit for current.
Response 9: Thank you for your reminder. We have checked and revised the data.
(Figure 7 displays the axial temperature distribution of molten region at f=30KHz and Icoil =70A、71A、73A 、75A、77A、79 A、81A, respectively.)
Point 10: In the subsection “4.1.3. Influence of current on axial temperature distribution”, the results for Icoil =71A, 73A, 75A, 77A, 79A, 81A are presented but the results for Icoil =70A should be also presented in Figure 6. Clarify why the values of Icoil were selected from 71A to 81A, whereas in subsection "4.1.1. Simulation Model validation" the studied range of Icoil was between 55A and 75A. The range of Icoil should be similar in all the subsections where the influence of Icoil is studied.
Response 10: The results for Icoil =70A have presented in Figure 7 (Figure 6 in the previous edition). And the reason why the values of Icoil were selected from 71A to 81A have been clarified.
(In order to prevent the temperature of the molten zone from being too low because of small current, the current in the range of 70-81A was selected for research . Figure 7 displays the axial temperature distribution of molten region at f=30KHz and Icoil =70A、71A、73A 、75A、77A、79 A、81A, respectively.)
Figure 7. Axial temperature distribution of molten region.
Point 11: In the subsection “4.1.4. Influence of frequency and current on molten region width”, the results for f = 30 KHz and Icoil = 71A, 75A, 81A are presented but the results for Icoil =70A should be also presented in Figure 7.
Response 11: We have supplemented the relevant data of Icoil =70A according to your suggestions.
Figure 7. The area of the valid molten region under different current. (a) Icoil = 70A; (b) Icoil = 71A; (c) Icoil = 75A; (d) Icoil = 81A.
Point 12: On page 8, lines 198-199, in the phrase “The relationship between the width of the molten region and frequency and current studied in the previous paper” should be mentioned the reference number of the previous paper of the authors. The zone refining rate and the number of passes of zone refining should be also specified in the actual manuscript submitted to Materials because in the actual formulation can be understood that these parameters refer to the previous paper of the authors.
Response 12: We have revised this sentence to prevent readers from misunderstandings, and the zone refining rate and the number of passes of zone refining have been specified in the actual manuscript.
(The relationship between the width of the molten region and frequency and current studied in section 4.1. In order to improve the efficiency of zone melting and ensure that the experimental equipment will not be damaged by excessive power current, the current frequency would be selected as 30KHz in in the experimental work. In the zone refining experiment, zone refining rate was 20mm/h, the current was set to 71A、75A and 81A respectively to control the molten zone width to 2cm, 4cm and 6cm. After ten passes of zone refining, the distribution of impurity concentration obtained after purification by ICP-AES.)
Point 13: On page 8, lines 203-204, in the phrase “Figure 9 and Figure 10 represent the concentration distribution of impurity Fe and Si in metal after zone refining under different conditions.” should be specified the conditions (e.g. current, frequency, melting temperature, including their values) instead expressing them in a general way.
Response 13: The conditions have already given in the previous paragraph.
(The relationship between the width of the molten region and frequency and current studied in section 4.1. In order to improve the efficiency of zone melting and ensure that the experimental equipment will not be damaged by excessive power current, the current frequency would be selected as 30KHz in in the experimental work. In the zone refining experiment, zone refining rate was 20mm/h, the current was set to 71A、75A and 81A respectively to control the molten zone width to 2cm, 4cm and 6cm. After ten passes of zone refining, the distribution of impurity concentration obtained after purification by ICP-AES.)
Point 14: On page 1, lines 34-38, “The equilibrium partition coefficient is defined as k0=CS/CL, which represents the solubility ratio of impurities in solid and liquid phase. During the zone refining process, the impurities of k0<1 will move to the end region of metal following the zone refining direction but impurities of k0 >1 will move along the opposite direction [8-10].”, whereas on page 9, line 227, it is mentioned “k < 1”. The notation for the equilibrium partition coefficient should be checked.
Response 14: The notation for the equilibrium partition coefficient have be checked. Thank you for reminding us.
(When k0 < 1, impurities enter the liquid phase through the melting interface, move from the beginning to the end, and finally move out of the molten region through the solidifica-tion interface.)
Point 15: On page 10, referring to “formula (4-1), (4-2)…”, should be replaced the word "formula" with "equation", and the numbering of the equations should be checked since the equations were noted as (1), (2), …
Response 15: We have replaced the word "formula" with "equation", sorry for that mistake.
(The limiting distribution equation has been given by Pfann [28-30], as shown in equation (9): )
Point 16: In Table 4, in column A for Fe and Si elements should be put as a superscript the number after each ×10 since it is an exponent. However, the results are presented only for the main impurities (Fe and Si) with a higher content (≥ 50 ppm). In the comments should be discussed why the other impurities that were shown in Table 3 (Cu, Cr, Mn, Zn, Ti, Pr, Ce, Co) with a lower content (< 50 ppm) were neglected.
Response 16: The format of the Table 4 have been checked and modified, and the reason why the other impurities that were shown in Table 3 (Cu, Cr, Mn, Zn, Ti, Pr, Ce, Co) with a lower content (< 50 ppm) were neglected have also been discussed.
(It can be seen from the Table 3 that Fe (787ppm) and Si (345ppm) were the main impurity in lanthanum, which need special attention in the experiment. The other impurities (Cu, Cr, Mn, Zn, Ti, Pr, Ce, Co) are close to limiting distribution due to its low content (< 50 ppm), which should be neglected, and the zone melting method has little effect on its purification.)
Table 4. The relationship between A and B and the width of molten zone.
|
|
Fe |
Si |
||
|
|
A |
B |
A |
B |
|
Z =2cm |
2.25×10-16 |
33.79 |
6.94×10-20 |
39.55 |
|
Z =4cm |
2.45×10-9 |
16.89 |
1.35×10-11 |
19.77 |
|
Z =6cm |
4.56×10-7 |
11.26 |
6.56×10-9 |
13.18 |
Point 17: On page 12, in section “5. Conclusions”, it was stated that “After zone refining, Fe and Si can be efficiently removed” but should be clarified the number of passes of zone refining and the purity, including the type and content of the remained impurities of the final material.
Response 17: Thank you very much for your valuable comment. We have carefully revised the conclusion.
(After 10 t
Dear Reviewer:
Thank you for your letter and comments concerning our manuscript entitled “Simulation and experimental research on purification of lan-thanum by zone refining” (ID: materials-1643748). Those comments are all valuable and very helpful for revising and improving our paper, as well as the important guiding significance to our researches. We have studied comments carefully and have made correction which we hope meet with approval. Revised portion are marked in red in the paper. The main corrections in the paper and the responds to the reviewer’s comments are as flowing:
Point 1: On page 1, line 30, in the phrase “However, due to its active chemical properties and low vapor pressure…” should be given some quantifiable data for the properties.
Response 1: Thank you for your reminder, and we have added the quantifiable vapor pressure data in introduction. Readers can find specific data in the reference [8].
(However, due to its active chemical properties and low vapor pressure(10-4Pa at 1500K), it is more difficult to improve its purity compared with other rare earth metals [8].)
Point 2: On page 3, referring to the heat conduction equation, radiation equation and Maxwell’s equations given in Table 2 should be provided proper references. Also, the constitutive terms of the equations should be briefly explained even the above equations are well known.
Response 2: Reference related to equations has been provided in this paper. And we have added explanations to all the symbols in this paper.
[26] Hou, Y. Q. Theoretical analysis and modeling of polysilicon preparation process by improved Siemens method. PhD thesis, Kunming University of Science and Technology, Kunming, 2013.
(The equilibrium partition coefficient (k0) is defined as k0=CS/CL, CS and CL represent the solubility ratio of impurities in solid phase and liquid phase, respectively.)
(When k0 < 1, impurities enter the liquid phase through the melting interface, move from the beginning to the end, and finally move out of the molten region through the solidification interface.)
(The heat conduction equation of the reactor can be expressed by equation (1):
(1)
K represents thermal conductivity of lanthanum, the relationship between thermal conductivity and temperature is as follows:
K=3.248059+0.03346795×T-2.471847×10-5×T2+9.863396×10×T3 (2)
q is defined as the formation heat per unit volume which can be calculated by equation (3):
(3)
represents voltage gradient per unit length, represents conductivity of lanthanum.
The heat caused by radiation can be calculated by equation (4):
(4)
Among them, is the Stefan-Boltzmann constant, 5.669×10−8 W/cm2 K−4, represents radiation coefficient of lanthanum, represents radiation coefficient of air, and are the temperature of lanthanum and air, respectively. and are the radius of lanthanum, are the width of air region.
The Maxwell equations can be expressed by equation (5-8):
(5)
(6)
(7)
(8)
E represents the electric field intensity, H is the magnetic induction intensity, is the dielectric permeability, represents the pulsation, is the permeability. The numerical values of the parameters used in the simulation were given in Table 2.)
Point 3: On page 3, lines 93-95, in the phrase “The numerical simulations shown in this paper was conducted using the commercial COMSOL Multiphysics finite element software, and the physical parameters of material can be found in database.” should be mentioned the physical parameters of material, including their values, that were used in the numerical simulations.
Response 3: I apologize for my negligence again and the physical parameters of material that were used in the numerical simulations have been metioned in Table 2.
(The numerical simulations shown in this paper was conducted using the commercial COMSOL Multiphysics finite element software, and the physical parameters of material can be found in Table2.)
Table 2. The numerical values of the parameters.
|
Parameter |
Value |
Unit |
|
1 |
H/m |
|
|
|
0.2 |
MS/cm |
|
0.64 |
1 |
|
|
0.5 |
1 |
|
|
1 |
1 |
Point 4: On page 4, lines 106-107, should be changed “0.01Mpa” with “0.01 MPa”. Referring to the phrase “The drive motor connected with the crucible was used to control the movement of the molten zone.” should be clarified how was controlled the movement of the molten zone, and which parameters and their values were involved in this step.
Response 4: To make it easier for readers to understand, we have described the experimental part in more detail. The related parameters and their values were involved in this step.
(To prevent oxidation, the experiment was carried out under vacuum conditions, and the vacuum of furnace was set at 0.01 MPa by vacuum pump. The coil was placed onto the crucible while keeping the coil position fixed. And the drive motor connected with the crucible was used to control the movement of the molten zone. As the crucible moves, the hearting part also changes. The movement rate was set at 20 mm/min.)
Point 5: On page 4, lines 111-113, should be specified the producer of the raw materials, the type, and purity of the molten salt, and electrolysis conditions, along with the processing parameters and their values to produce a round bar with a length of L = 20 cm and a radius of 3 cm. The round bar should be also analyzed in terms of physical properties and the results should be also presented in the manuscript to compare the input data from the numerical simulations with the ones from the experimental works.
Response 5: The materials used for the experiment were purchased directly from other companies.
The type of the molten salt、purity、electrolysis conditions and processing parameters are confidential and therefore cannot be available. There is also a lack of effective measurement methods for the physical properties. In addition, there is a lack of effective measurement methods for us to analyzed the round bar.
(The raw materials used for the experiment were lanthanum produced by molten salt electrolysis from LeShan Grinm Advanced materials, which were processed into a round bar with a length of L = 20cm and a radius of 3cm and placed in the crucible.)
Point 6: On page 4, lines 113-114, in the phrase “Inductively coupled plasma atomic emission spectrometry was used to analyze lanthanum composition…” should be mentioned the type of the spectrometer and the measurement conditions. Should be also clarified the zone from which were taken samples for measurements, and how was studied the material homogeneity.
Response 6: The whole composition analysis was entrusted to relevant analysis institutions. We don't know the specific measurement conditions and types, which is not the focus of this paper. And the molten salt electrolysis is regarded as the most uniform method for making metals. Generally, the uniformity of materials produced by molten salt electrolysis does not need to be considered. But your question is very suggestive and deserves our attention in the future.
Point 7: On page 4, lines 126-127, should be given a reference for the equation showing the average relative error, and the number of considered temperatures (n) used to determine the average value should be also specified even in Figure 3 can be seen 5 temperatures for both calculated and experimental temperatures.
Response 7: The relevant reference and the number of considered temperatures have been added.
(By comparing 5 temperatures for both calculated and experimental temperatures, the average relative error () was used to evaluate the proximity between the calculated temperature and the experimental temperature [27].)
- Ya, D. L.; Gang, X. A CFD model for gas uniform distribution in turbulent flow for the production of titanium pigment in chloride process. Chinese Journal of Chemical Engineering 2016, 24(06), 749-756.
Point 8: On page 5, in subsection “4.1.2. Influence of frequency on radial temperature distribution” are given the results for the temperature distribution of the zone refining reactor at Icoil = 75 A and f = 30 KHz. I recommend you to show the results in Figure 4 and Figure 5 with adequate comments for Icoil = 70 A and f = 30 KHz, where the calculated temperature was in very good agreement with the experimental temperature (Figure 3). Figure 5 (Radial temperature distribution of molten region) presents the simulation results for f = 20 KHz, f = 30 KHz, and f = 40 KHz, but in the experimental work was selected f = 30 KHz. In this respect, should be clarified the selection of the frequency f = 30 KHz in the experimental work.
Response 8: Thank you for your careful reading and valuable suggestions. The data in Figure 4 has been replaced with Icoil = 70 A and f = 30 KHz. And the reason for selecting f = 30 KHz has also clarified. The reason why 75A was selected instead of 70A in Figure. 5 is that when the current was 70A and the frequency was 20kHz, the temperature of the molten zone was low and there was an unmelted area, which is not conducive to the migration of subsequent impurities.
(Figure 4 represents the temperature distribution of the zone refining reactor at Icoil = 70A and f = 30KHz. As shown in the figure, the surface temperature of the melting zone was 1020°C, where the calculated temperature was in very good agreement with the experimental temperature.)
Figure 4. The temperature profile of zone melting reactor. (a) 3D; (b) 2D.
(In order to improve the efficiency of zone melting and ensure that the experimental equipment will not be damaged by excessive power current, the current frequency would be selected as 30kHz in in the experimental work.)
Point 9: On page 6, line 161, should be checked “f = 75A” since f is the abbreviation for frequency, and A is the measurement unit for current.
Response 9: Thank you for your reminder. We have checked and revised the data.
(Figure 7 displays the axial temperature distribution of molten region at f=30KHz and Icoil =70A、71A、73A 、75A、77A、79 A、81A, respectively.)
Point 10: In the subsection “4.1.3. Influence of current on axial temperature distribution”, the results for Icoil =71A, 73A, 75A, 77A, 79A, 81A are presented but the results for Icoil =70A should be also presented in Figure 6. Clarify why the values of Icoil were selected from 71A to 81A, whereas in subsection "4.1.1. Simulation Model validation" the studied range of Icoil was between 55A and 75A. The range of Icoil should be similar in all the subsections where the influence of Icoil is studied.
Response 10: The results for Icoil =70A have presented in Figure 7 (Figure 6 in the previous edition). And the reason why the values of Icoil were selected from 71A to 81A have been clarified.
(In order to prevent the temperature of the molten zone from being too low because of small current, the current in the range of 70-81A was selected for research . Figure 7 displays the axial temperature distribution of molten region at f=30KHz and Icoil =70A、71A、73A 、75A、77A、79 A、81A, respectively.)
Figure 7. Axial temperature distribution of molten region.
Point 11: In the subsection “4.1.4. Influence of frequency and current on molten region width”, the results for f = 30 KHz and Icoil = 71A, 75A, 81A are presented but the results for Icoil =70A should be also presented in Figure 7.
Response 11: We have supplemented the relevant data of Icoil =70A according to your suggestions.
Figure 7. The area of the valid molten region under different current. (a) Icoil = 70A; (b) Icoil = 71A; (c) Icoil = 75A; (d) Icoil = 81A.
Point 12: On page 8, lines 198-199, in the phrase “The relationship between the width of the molten region and frequency and current studied in the previous paper” should be mentioned the reference number of the previous paper of the authors. The zone refining rate and the number of passes of zone refining should be also specified in the actual manuscript submitted to Materials because in the actual formulation can be understood that these parameters refer to the previous paper of the authors.
Response 12: We have revised this sentence to prevent readers from misunderstandings, and the zone refining rate and the number of passes of zone refining have been specified in the actual manuscript.
(The relationship between the width of the molten region and frequency and current studied in section 4.1. In order to improve the efficiency of zone melting and ensure that the experimental equipment will not be damaged by excessive power current, the current frequency would be selected as 30KHz in in the experimental work. In the zone refining experiment, zone refining rate was 20mm/h, the current was set to 71A、75A and 81A respectively to control the molten zone width to 2cm, 4cm and 6cm. After ten passes of zone refining, the distribution of impurity concentration obtained after purification by ICP-AES.)
Point 13: On page 8, lines 203-204, in the phrase “Figure 9 and Figure 10 represent the concentration distribution of impurity Fe and Si in metal after zone refining under different conditions.” should be specified the conditions (e.g. current, frequency, melting temperature, including their values) instead expressing them in a general way.
Response 13: The conditions have already given in the previous paragraph.
(The relationship between the width of the molten region and frequency and current studied in section 4.1. In order to improve the efficiency of zone melting and ensure that the experimental equipment will not be damaged by excessive power current, the current frequency would be selected as 30KHz in in the experimental work. In the zone refining experiment, zone refining rate was 20mm/h, the current was set to 71A、75A and 81A respectively to control the molten zone width to 2cm, 4cm and 6cm. After ten passes of zone refining, the distribution of impurity concentration obtained after purification by ICP-AES.)
Point 14: On page 1, lines 34-38, “The equilibrium partition coefficient is defined as k0=CS/CL, which represents the solubility ratio of impurities in solid and liquid phase. During the zone refining process, the impurities of k0<1 will move to the end region of metal following the zone refining direction but impurities of k0 >1 will move along the opposite direction [8-10].”, whereas on page 9, line 227, it is mentioned “k < 1”. The notation for the equilibrium partition coefficient should be checked.
Response 14: The notation for the equilibrium partition coefficient have be checked. Thank you for reminding us.
(When k0 < 1, impurities enter the liquid phase through the melting interface, move from the beginning to the end, and finally move out of the molten region through the solidifica-tion interface.)
Point 15: On page 10, referring to “formula (4-1), (4-2)…”, should be replaced the word "formula" with "equation", and the numbering of the equations should be checked since the equations were noted as (1), (2), …
Response 15: We have replaced the word "formula" with "equation", sorry for that mistake.
(The limiting distribution equation has been given by Pfann [28-30], as shown in equation (9): )
Point 16: In Table 4, in column A for Fe and Si elements should be put as a superscript the number after each ×10 since it is an exponent. However, the results are presented only for the main impurities (Fe and Si) with a higher content (≥ 50 ppm). In the comments should be discussed why the other impurities that were shown in Table 3 (Cu, Cr, Mn, Zn, Ti, Pr, Ce, Co) with a lower content (< 50 ppm) were neglected.
Response 16: The format of the Table 4 have been checked and modified, and the reason why the other impurities that were shown in Table 3 (Cu, Cr, Mn, Zn, Ti, Pr, Ce, Co) with a lower content (< 50 ppm) were neglected have also been discussed.
(It can be seen from the Table 3 that Fe (787ppm) and Si (345ppm) were the main impurity in lanthanum, which need special attention in the experiment. The other impurities (Cu, Cr, Mn, Zn, Ti, Pr, Ce, Co) are close to limiting distribution due to its low content (< 50 ppm), which should be neglected, and the zone melting method has little effect on its purification.)
Table 4. The relationship between A and B and the width of molten zone.
|
|
Fe |
Si |
||
|
|
A |
B |
A |
B |
|
Z =2cm |
2.25×10-16 |
33.79 |
6.94×10-20 |
39.55 |
|
Z =4cm |
2.45×10-9 |
16.89 |
1.35×10-11 |
19.77 |
|
Z =6cm |
4.56×10-7 |
11.26 |
6.56×10-9 |
13.18 |
Point 17: On page 12, in section “5. Conclusions”, it was stated that “After zone refining, Fe and Si can be efficiently removed” but should be clarified the number of passes of zone refining and the purity, including the type and content of the remained impurities of the final material.
Response 17: Thank you very much for your valuable comment. We have carefully revised the conclusion.
(After 10 times zone refining and holding the width of molten at 2cm, the concentration of Fe and Si in the beginning were 152ppm and 45ppm respectively, which were much lower than that in the raw material, indicating that Fe and Si can be efficiently removed.)
Special thanks to you for your good comments.
imes zone refining and holding the width of molten at 2cm, the concentration of Fe and Si in the beginning were 152ppm and 45ppm respectively, which were much lower than that in the raw material, indicating that Fe and Si can be efficiently removed.)
Special thanks to you for your good comments.

Reviewer 3 Report
The authors have been modeled the purification of lanthanum by the finite element method. The output is compared with experimental results. manuscript is well organized and the result can be used in the practical study. The following items should be corrected in the manuscript.
- In the last paragraph of the introduction, it is necessary to present the innovation of the study compared to the previous publications, especially the presented paper in the introduction of the manuscript.
- The title of the manuscript can be changed to better reflect the study as follows: Simulation of lanthanum purification using finite element method.
- It is necessary to specify the dimensions of the model or modeling in Figure (1)
- Since the modeling of this study is based on the finite element method, it is necessary to add some paper on the application of this method in engineering to the introductory part of the article. for this purpose this reviewer suggests the following papers:
Stochastic analysis of rainfall-induced slope instability and steady-state seepage flow using random finite-element method, (2019)
Finite element analysis of impact damage response of composite motorcycle safety helmets, (2002)
- The presented equations in Table 2 should be taken out of the table and all their parameters are defined.
- Section 4 does not explain element size and boundary. As the authors know, the size of the elements is very influential on the output solutions. Therefore it is necessary to perform a sensitivity analysis to measure the size of the elements to obtain the appropriate element size.
- According to Figure 7, The dimensions of the finite element model must be such that the temperature zone reaches near zero, which is not the case in these figures. Modeling dimensions need to be larger
- Conclusion No. 3 is obvious and should be omitted.
Author Response
Dear Reviewer:
Thank you for your letter and comments concerning our manuscript entitled “Simulation and experimental research on purification of lan-thanum by zone refining” (ID: materials-1643748). Those comments are all valuable and very helpful for revising and improving our paper, as well as the important guiding significance to our researches. We have studied comments carefully and have made correction which we hope meet with approval. Revised portion are marked in red in the paper. The main corrections in the paper and the responds to the reviewer’s comments are as flowing:
Point 1: In the last paragraph of the introduction, it is necessary to present the innovation of the study compared to the previous publications, especially the presented paper in the introduction of the manuscript.
Response 1: Thank you for your valuable and thoughtful comments. The innovation of the study compared to the previous publications have presented in the last paragraph of the introduction.
(The present study differs from prior studies in important ways. In this study, we report, for the first, the relationship between the width of the molten region and AC current. It also reveals the impurity distribution under different widths of molten zone which has a great guiding significance for the actual melting process.)
Point 2: The title of the manuscript can be changed to better reflect the study as follows: Simulation of lanthanum purification using finite element method.
Response 2: After careful consideration, we decided to adopt your suggestion and change the title of the manuscript to “Simulation of lanthanum purification using finite element method”.
Point 3: It is necessary to specify the dimensions of the model or modeling in Figure (1)
Response 3: The specific geometric dimensions have been indicated in the Figure1. It can also be found in Table 1.
Figure 1. Geometrical model of zone refining reactor.
Table 1. Dimensions and boundary conditions of zone refining reactor.
|
Region |
Items |
Data |
|
Water-cooled coil |
Outer diameter |
7 mm |
|
Inner diameter |
5 mm |
|
|
Air region |
Width |
150 mm |
|
Length |
400 mm |
|
|
Lanthanum |
Width |
30 mm |
|
Length |
200 mm |
|
|
Boundary conditions |
Temperature of cooling water |
283 k |
|
Temperature of reactor wall |
293 k |
|
|
Cooling water mass flow rate |
1 kg/min |
|
|
Emissivity of metal |
0.5 |
Point 4: Since the modeling of this study is based on the finite element method, it is necessary to add some paper on the application of this method in engineering to the introductory part of the article. for this purpose this reviewer suggests the following papers:
Stochastic analysis of rainfall-induced slope instability and steady-state seepage flow using random finite-element method, (2019)
Finite element analysis of impact damage response of composite motorcycle safety helmets, (2002)
Response 4: The recent numerical work have been added in the main text.
A significant amount of work has also been done on the finite element method [18-19]. Cheung combined a numerical model and genetic algorithm to establish an optimized melting zone length model to achieve maximum purification efficiency [20]. Tan Yang reported that semicircular canals was more suitable than square grooves for the purifica-tion of zone refining by combining finite element simulation data and experimental re-sults analysis [21]. Chen applied the finite element analysis method to simulate flow field distribution of the refining region [22].
[18]. Johari, A.; Talebi, A. Stochastic Analysis of Rainfall-Induced Slope Instability and Steady-State Seepage Flow Using Random Finite-Element Method. International Journal of Geomechanics 2019, 19(8).
[19]. Kostopoulos, V.; Markopoulos, Y. P.; Giannopoulos, G. et al. Finite element analysis of impact damage response of composite motorcycle safety helmets. Composites Part B Engineering 2002, 33(2):99-107.
Point 5: The presented equations in Table 2 should be taken out of the table and all their parameters are defined.
Response 5: Thank you for reminding me, and the presented equations in Table 2 have been taken out of the table.
(The heat conduction equation of the reactor can be expressed by equation (1):
(1)
K represents thermal conductivity of lanthanum, the relationship between thermal conductivity and temperature is as follows:
K=3.248059+0.03346795×T-2.471847×10-5×T2+9.863396×10×T3 (2)
q is defined as the formation heat per unit volume which can be calculated by equation (3):
(3)
represents voltage gradient per unit length, represents conductivity of lanthanum.
The heat caused by radiation can be calculated by equation (4):
(4)
Among them, is the Stefan-Boltzmann constant, 5.669×10−8 W/cm2 K−4, represents radiation coefficient of lanthanum, represents radiation coefficient of air, and are the temperature of lanthanum and air, respectively. and are the radius of lanthanum, are the width of air region.
The Maxwell equations can be expressed by equation (5-8):
(5)
(6)
(7)
(8)
E represents the electric field intensity, H is the magnetic induction intensity, is the dielectric permeability, represents the pulsation, is the permeability. The numerical values of the parameters used in the simulation were given in Table 2.)
Point 6: Section 4 does not explain element size and boundary. As the authors know, the size of the elements is very influential on the output solutions. Therefore it is necessary to perform a sensitivity analysis to measure the size of the elements to obtain the appropriate element size.
Response 6: Information about the element size has given in 2.3 Numerical method, and boundary conditions can also be found in Table1
(2.3 Numerical method
The numerical simulations shown in this paper was conducted using the commercial COMSOL Multiphysics finite element software, and the physical parameters of mate-rial can be found in database. The grid division of computational domain have a pro-found effect on the precision of results. Considering the symmetry of reactor and the limi-tation of the grid node structure, the triangular mesh was utilized for computational mesh generation, as shown in Figure 2. After meshing, the total number of grids was 6283, and the mesh was generated with a maximum element size of 1×10-2 mm and minimum of 7.86×10-5 mm. Computations of skewness (maximum 0.61) and orthogonal quality (min-imum 0.32) indicated an acceptable quality of the triangular in the mesh.)
Table 1. Dimensions and boundary conditions of zone refining reactor.
|
Region |
Items |
Data |
|
Water-cooled coil |
Outer diameter |
7 mm |
|
Inner diameter |
5 mm |
|
|
Air region |
Width |
150 mm |
|
Length |
400 mm |
|
|
Lanthanum |
Width |
30 mm |
|
Length |
200 mm |
|
|
Boundary conditions |
Temperature of cooling water |
283 k |
|
Temperature of reactor wall |
293 k |
|
|
Cooling water mass flow rate |
1 kg/min |
|
|
Emissivity of metal |
0.5 |
Point 7: According to Figure 7, The dimensions of the finite element model must be such that the temperature zone reaches near zero, which is not the case in these figures. Modeling dimensions need to be larger
Response 7:
In order to observe the valid molten region more clearly, we have processed the temperature distribution graph to show the area with temperature greater than 920℃. As shown in Figure(a), it is harder to measure the width of the molten zone than Figure(b). That is the reason why the temperature zone is not close to zero. In order to ensure the accuracy of the simulation results, the modeling dimensions was obtained by measuring the actual reactor.
(a)Raw date(10.1℃~1300℃) (b) Processed data (920℃~1300℃)
Point 8: Conclusion No. 3 is obvious and should be omitted.
Response 8: According to your suggestion, Conclusion No. 3 has be deleted.
Special thanks to you for your good comments.

Round 2
Reviewer 2 Report
The manuscript is recommended for publishing in Materials since the authors performed a satisfactory revision.